# Covichem: A biochemical severity risk score of COVID-19 upon hospital admission

**Marie-Lise Bats[1,2], Benoit Rucheton[3], Tara Fleur[1], Arthur Orieux[4], Clément Chemin[1], Sébastien Rubin[2,5], Brigitte Colombies[1], Arnaud Desclaux[6], Claire Rivoisy[7], Etienne Mériglier[7], Etienne Rivière[8], Alexandre Boyer[4], Didier Gruson[4], Isabelle Pellegrin[9,10], Pascale Trimoulet[11,12], Isabelle Garrigue[11,12], Rana Alkouri[3], Charles Dupin[13,14], François Moreau-Gaudry[1,13], Aurélie Bedel[1,13], Sandrine Dabernat[1,13]***

1 Department of Biochemistry, Pellegrin Hospital, University Hospital of Bordeaux, Bordeaux, France, 2 Inserm U1034, Biology of Cardiovascular Diseases, Pessac, France, 3 Department of Metabolic Biochemistry, DMU BioGeM, AP-HP Sorbonne University, University Hospitals of Pitié-Salpêtrière - Charles Foix, Paris, France, 4 Intensive Care Medicine Unit, Pellegrin and Saint-André Hospitals, University Hospital of Bordeaux, Bordeaux, France, 5 Department of Nephrology-Transplantation-Dialysis, University Hospital of Bordeaux, Bordeaux, France, 6 Department of Infectious and Tropical Diseases, Pellegrin Hospital, University Hospital of Bordeaux, Bordeaux, France, 7 Department of Internal Medicine, Saint-André Hospital, University Hospital of Bordeaux, Bordeaux, France, 8 Department of Internal Medicine, Haut-Lévêque Hospital, University Hospital of Bordeaux, Bordeaux, France, 9 Biological Resources Center, University Hospital of Bordeaux, Bordeaux, France, 10 Laboratory of Immunology and Immunogenetics, CHU Bordeaux, Bordeaux, France, 11 Department of Virology, Pellegrin Hospital, University Hospital of Bordeaux, Bordeaux, France, 12 CNRS-UMR 5234, University of Bordeaux, Bordeaux, France, 13 Inserm U1035, Bordeaux, Bordeaux, France, 14 Department of Radiotherapy, University Hospital of Bordeaux, Bordeaux, France

* sandrine.dabernat@chu-bordeaux.fr

**Data Availability Statement:** All relevant data are within the manuscript and its Supporting information files.

**Funding:** The author(s) received no specific funding for this work.

## Abstract

Clinical and laboratory predictors of COVID-19 severity are now well described and combined to propose mortality or severity scores. However, they all necessitate saturable equipment such as scanners, or procedures difficult to implement such as blood gas measures. To provide an easy and fast COVID-19 severity risk score upon hospital admission, and keeping in mind the above limits, we sought for a scoring system needing limited invasive data such as a simple blood test and co-morbidity assessment by anamnesis. A retrospective study of 303 patients (203 from Bordeaux University hospital and an external independent cohort of 100 patients from Paris Pitié-Salpêtrière hospital) collected clinical and biochemical parameters at admission. Using stepwise model selection by Akaike Information Criterion (AIC), we built the severity score Covichem. Among 26 tested variables, 7: obesity, cardiovascular conditions, plasma sodium, albumin, ferritin, LDH and CK were the independent predictors of severity used in Covichem (accuracy 0.87, AUROC 0.91). Accuracy was 0.92 in the external validation cohort (89% sensitivity and 95% specificity). Covichem score could be useful as a rapid, costless and easy to implement severity assessment tool during acute COVID-19 pandemic waves.

**Competing interests:** The authors have declared that no competing interests exist.

# Introduction

About 14% of patients infected by severe acute respiratory syndrome coronavirus 2 (SARS-CoV-2) need hospitalization and oxygen support and 5% require admission to an intensive care unit [1]. Among the needed tools to fight against COVID-19, the early identification of clinical and laboratory predictors of disease severity retained special attention early on in the immediate evaluation of hospital resources [2, 3]. As in April 2020, blood routine parameters were found to provide important information for the severity of disease since they were significantly different between non-severe and severe types of COVID-19 patients. However only few parameters (CRP, D-Dimer and albumin) showed high consistency between studies [4, 5]. A meta-analysis assessed the value of mortality and severity scores, published in 30 studies [6]. Among them, almost 3 out of 4 assessed mortality risk and half (16/30) the severity risk. Among the reported severity risks, 2 out of 3 (12) are actually non-peer-reviewed studies, shared online by the authors on dedicated platforms. Only 4 peer-reviewed articles report scores associated to severity; among them two scores used blood markers to predict disease severity at hospital admission [7, 8]. Overall, the other available publications are only descriptive of routine biochemical parameters and observed differences were somewhat expected. While albumin was found inversely correlated and lactate dehydrogenase (LDH) and C-reactive protein (CRP) positively correlated with Murray scores documenting the severity of lung injury [9], the combination of these parameters upon hospital admission was not tested as a predictive factor of COVID-19 severity.

In this study, we tested whether a limited number of biochemical parameters values at the time of admission could provide a COVID-19 severity score.

# Materials and methods

According to recent recommendations [6], this study adheres to the TRIPOD (transparent reporting of multivariable prediction model for individual prognosis or diagnosis) reporting guideline [10].

## Participants and source of data

The retrospective discovery consecutive cohort included patients hospitalized from March 4, to May 7, 2020 in the departments of infectious diseases, internal medicine or intensive care units (ICU) of the University Hospital of Bordeaux, France. According to French law and the French Data Protection Authority, the handling of these data for research purposes was declared to the Data Protection Officer of the University Hospital of Bordeaux and AP-HP (Assistance Publique-Hôpitaux de Paris). The study was approved by the Institutional Review Board and Ethics Committee, which waived the requirement for informed consent (declaration number GP-CE-2020-20).

Participants (n = 222) were enrolled if they had a positive SARS-CoV-2 polymerase chain reaction (from nasopharyngeal swab test) and/or typical computed lung tomography images associated with a high clinical probability of COVID-19, including the usual symptoms, among them: dry cough, fever, chills, fatigue, dyspnea, chest pain, myalgia, diarrhea, anosmia and ageusia [11, 12].

Patients' demographic data (age, sex, body mass index (BMI)), clinical features (date and COVID-19 symptoms, hospitalization duration, chronic comorbidities), and laboratory parameters were routinely collected during their hospital stay in dedicated electronic health records (DXCare® and Metavision® softwares). Biochemical data on natremia, kaliemia, total proteins, albumin, CRP, alkaline phosphatase (ALP), alanine transaminase (ALT), aspartate aminotransferase (AST), ferritin, creatine kinase (CK) and LDH were reviewed within the first

days after admission (average of 1.5 days). According to the 1st exclusion rule related to patient level completeness, patients who had more than 20% missing values were excluded (n = 19 patients). In total, 203 patients were included in the study (S1 Fig).

The retrospective consecutive validation cohort included 100 confirmed COVID-19 patients (45 severe and 55 non-severe), according to the same criteria as above, admitted to Pitié-Salpêtrière hospital in Paris, France, in internal medicine units or ICUs from March 31st to April 4th 2020. Data were collected from Orbis® software. All biochemical results were obtained within the first 24h after hospital admission.

## Outcome

The study participants were divided into two groups: severe and non-severe patients. The severity was defined with the following criteria: arterial oxygen saturation (SaO2) less than 90% on room air or need of $\geq$ 4 L/min oxygen therapy (O2) to obtain a SaO2 $\geq$ 94% [13]. Patients were considered severe if one of these criteria was present at the admission or occurred during their hospital stay. Patients with acute respiratory distress syndrome at the admission or/and directly admitted to the ICU were also included. All the patients without the cited severe signs were included in the non-severe group.

## Severity prediction

**Finding significant severity predictors.** Correlation analyses evaluated the strength of relationship between two variables, including the severity. Twenty-nine variables were tested: length of hospitalization stay, age, sex, obesity, BMI, hypertension, diabetes, smoking, dyslipidemia, cardiovascular, infectious, inflammatory, respiratory, renal, liver diseases, cancer, viral load E gene and ORF1, natremia, kaliemia, total proteins, albumin, CRP, ferritin, AST, ALT, ALP, CK and LDH. Pearson's correlation coefficients represented the degree of linear association between COVID-19 severity and each of the 29 variables.

Receiver operating characteristic (ROC) curves measured the predictive value of COVID-19 severity for single clinical or biological variables.

**Missing data.** According to the 2nd exclusion rule related to variable level completeness, variables with more than 40% missing data were excluded to build the predictive model (e.g. BMI). All the variables were available for 118 patients. Missing values in the population data were imputed using random forests [14].

**Model construction and validation.** In order to work with an explainable predictive model, a multivariate logistic regression was fitted with 27 variables (excluding hospitalization duration and BMI).

We performed an 80% random split for the training set with the caret R package. The random sampling was done within the 2 levels of severity in an attempt to balance the class distributions within the splits. In total, 40 patients were randomly selected from the total population as test set and the 163 remaining patients were designated as the training set (S1 Fig).

Significant predictors were selected in the training group by performing stepwise model selection by Akaike Information Criterion (AIC).

From the estimates, we computed the effects for each predictor, summed them up and by applying a logistic transformation, we derived a severity score also called Covichem score, with the following logistic equation:

$$\text{Covichem score} = 1/\left(1 + \exp\left[-\left(\beta_0 + \sum\nolimits_{j=1}^{k} \beta_j X_j\right)\right]\right)$$

$\beta0$ is the intercept and $\beta j$ are the estimates for each predictor $Xj$. A score > 0.5 was defined by AUROC as the cut-off for severity.

To assess how the prediction will generalize to an independent new data set, the accuracy of the model was estimated by a resampling Leave-One-Out Cross-Validation technique.

Statistical performance for the Covichem score were evaluated in both training and test sets by calculating accuracy, sensitivity, specificity, negative and positive predictive values (NPV and PPV) and area under ROC curve (AUROC).

### External validation

We used the Covichem score to predict the severity risk in patients from Pitié-Salpêtrière Hospital (Paris). The accuracy, sensitivity, specificity, NPV and PPV were computed to evaluate the model performance.

### Biochemical assays

Biochemical parameters in Bordeaux university hospitals were measured on plasmas collected on Vacutainer® Barricor tubes (Becton Dickinson, Le-Pont-de-Claix, France), using Architect analyzers (Abbott Diagnostics, Rungis, France). The following analytical methods were used: indirect potentiometry for plasma sodium and potassium, colorimetry for total proteins and albumin (bromocresol purple method), enzymatic method for ALP, ALT, AST, LDH and CK, immunoturbidimetry for CRP and immunochemiluminescence for ferritin. Exploration of kidney function was not included in data recovery because published data showed that urea and creatinine remained in normal ranges.

In Pitié-Salpêtrière hospital, plasmas were collected on Vacutainer® PST Lithium Heparinate tubes (Becton Dickinson, Le-Pont-de-Claix, France). Biochemical parameters were measured on Cobas c 8000 module analyzers (Roche Diagnostics, Meylan, France), using the following analytical methods: indirect potentiometry for plasma sodium, enzymatic method for LDH and CK and immunoturbidimetry for albumin (Diagam) and ferritin.

### Statistical analysis

Continuous and discrete variables were expressed as median (25th, 75th percentile) and absolute (relative) frequencies of patients, respectively. To compare the differences between severe and non-severe patients, we used Wilcoxon-Mann-Whitney U test for quantitative variables and Chi-squared test for categorical variables. A value of double-sided $p < 0.05$ was considered statistically significant.

All analyses were performed using R 3.6.3 (R Foundation for Statistical Computing, Vienna, Austria) or GraphPad Prism 5.0 (GraphPad Software, Inc., San Diego, CA, USA). The model development and validation were implemented using caret R package.

## Results

### Baseline characteristics of the population

Patients hospitalized in Bordeaux university hospital were included from March 4th to April 27th, 2020 (S1 Fig). They came to the hospital for suspicion of SARS-CoV-2 infection. Among the 222 enrolled patients, 203 were eligible, with >80% of the necessary clinical and biochemical available data. Ninety one percent of eligible patients were positive for virus detection by RT-QPCR. The negative group presented common infection symptoms, including dry cough, fever, sore throat and typical lung lesions on the chest CT-scan. COVID-19 was severe for 97 patients (48%, Table 1). Sixty-eight patients (33%) were admitted in the ICU, either directly at

**Table 1. Baseline characteristics of the COVID-19 patient cohort.**

| Variable | All, n = 203 | Non-severe, n = 106 (52%) | Severe, n = 97 (48%) | p-value |
|---|---|---|---|---|
| **Demographic** | | | | |
| Age (years) | 62 (51, 74) | 59 (47.8, 72.3) | 67 (58.5, 76) | 0.0011 |
| Male sex, n (%) | 107 (53%) | 48 (44%) | 59 (62%) | 0.0176 |
| **Chronology of disease** | | | | |
| Onset time (days) | 6 (3, 8) | 6 (3, 9) | 6 (3, 8) | 0.7721 |
| Duration of hospitalization (days) | 8.5 (5–19) | 6 (2–9) | 22 (13–33) | <0.001 |
| **Cardiovascular risk factors** | | | | |
| Diabetes, n (%) | 39 (19%) | 16 (15%) | 23 (24%) | 0.1277 |
| Dyslipidemia, n (%) | 45 (22%) | 22 (20%) | 23 (24%) | 0.6216 |
| Hypertension, n (%) | 81 (40%) | 32 (30%) | 49 (52%) | 0.0022 |
| Smoking, n (%) | 39 (19%) | 17 (16%) | 22 (23%) | 0.2437 |
| Obesity, n (%) | 47 (23%) | 13 (12%) | 34 (36%) | <0.001 |
| BMI (kg/m$^2$) | 27.3 (24.2, 31.8) | 26.1 (22.4, 29.1) | 29.4 (25.6, 32.9) | 0.0012 |
| **Others comorbidities** | | | | |
| Cardiovascular disease, n (%) | 66 (33%) | 23 (21%) | 43 (45%) | <0.001 |
| Cancer, n (%) | 35 (17%) | 18 (17%) | 17 (18%) | 0.9091 |
| Infectious disease, n (%) | 8 (4%) | 6 (6%) | 2 (2%) | 0.3846 |
| Inflammatory disease, n (%) | 23 (11%) | 13 (12%) | 10 (11%) | 0.9498 |
| Liver disease, n (%) | 6 (3%) | 3 (3%) | 3 (3%) | 1.0000 |
| Renal disease, n (%) | 10 (5%) | 4 (4%) | 6 (6%) | 0.5670 |
| Respiratory disease, n (%) | 44 (22%) | 24 (22%) | 20 (21%) | 1.0000 |
| **SARS-CoV-2 viral load** | | | | |
| *ORF1* (Ct value) | 27.5 (23.2, 31.6) | 27.9 (22.9, 31.8) | 27.3 (23.5, 31.1) | 0.7558 |
| *E-gene* (Ct value) | 28.8 (24, 33.8) | 29.3 (23.8, 34) | 27.9 (24, 32.7) | 0.3727 |
| **Biochemical parameters** | | | | |
| Natremia, mmol/L | 138 (135, 140) | 139 (136, 140) | 136 (134, 139) | 0.0047 |
| Kaliemia, mmol/L | 3.91 (3.63, 4.15) | 3.90 (3.67, 4.08) | 3.91 (3.57, 4.22) | 0.7493 |
| Total proteins, g/L | 72 (67, 76) | 73 (68, 77) | 71 (66, 75) | 0.0090 |
| Albumin, g/L | 28.4 (23.6, 33.3) | 32.2 (28, 36.4) | 24.6 (19.2, 28.4) | <0.001 |
| CRP, mg/L | 83.9 (32.9, 163.3) | 57.1 (11, 108.9) | 128.7 (65.7, 199.5) | <0.001 |
| ALP, U/L | 68 (58, 87) | 67 (58, 79) | 70 (59, 102) | 0.1111 |
| AST, U/L | 41 (30, 58) | 36 (29, 46) | 50 (35, 70) | <0.001 |
| ALT, U/L | 28 (18, 45) | 26 (17, 41) | 30 (22, 48) | 0.0264 |
| Ferritin, ng/mL | 581 (294, 1139) | 367 (169, 708) | 973 (516, 2155) | <0.001 |
| LDH, U/L | 339 (270, 452) | 284 (231, 366) | 392 (332, 516) | <0.001 |
| CK, U/L | 85 (48, 201) | 66 (40, 120) | 118 (60, 305) | <0.001 |

Onset time corresponds to the days between the onset of symptoms and the admission to hospital. Cardiovascular diseases include coronary artery diseases such as angina and myocardial infarction, heart failure, cardiomyopathy, abnormal heart rhythms, valvular heart disease, aortic aneurysms, heart transplant, peripheral artery disease, thromboembolic disease, venous thrombosis and stroke. Continuous and discrete variables are presented as median (25th, 75th percentile) and number (%) of patients and analyzed using Wilcoxon-Mann-Whitney U test and Chi-squared test, respectively. ALP, Alkaline Phosphatase; ALT, Alanine Aminotransferase; AST, Aspartate Aminotransferase; BMI, Body Mass Index; CK, Creatine Kinase; CRP, C-reactive protein; Ct, Cycle threshold; LDH, Lactate Dehydrogenase.

the admission or after a median hospital stay of 3 days. Mortality rate was 12% (25/203 patients) and occurred mostly in the group of patients with severe COVID-19 (24/25 patients). The median age was 62 years and sex ratio was 1.11 (M/F), both parameters being associated to disease severity (p = 0.0011 and p = 0.017, respectively, Table 1). Comorbidities associated

to severity were obesity, high blood pressure and cardiovascular conditions distinct from high blood pressure (Table 1).

Median time between disease symptoms and hospitalization was the same in both severe and non-severe COVID-19 patients (Table 1, 6 days). The median hospitalization stay length was almost 4 times longer for the severe group (22 days versus 6 days, Table 1). Interestingly, 83% of non-severe patients stayed less than 10 days whereas 80% of severe patients stayed more than 10 days.

We examined whether differences were seen among the biochemical analytes linked to cytolysis and/or liver function (LDH, ASAT, ALAT, CK, and PAL), inflammation (CRP, ferritin) and standard biochemical analytes (total proteins, albumin, sodium, potassium). Median values (Table 1) and Pearson's coefficients (S1 Table) were obtained according to severity. A correlogram identified parameters linearly correlated to severity and examined clinical data relevant to severity (Fig 1). Density plots describe the distribution of continuous variables in both groups (S2 Fig). Interestingly, the continuous variables best fitting severity were albumin (a drop of almost 25%, ρ = 0.55), LDH (around 1.5-fold increase, ρ = 0.40), ferritin and CRP (almost 3-fold increase, ρ = 0.38 and 0.34, respectively). The severe group counted 3 times more obese people and 2 times more patients with cardiovascular conditions other than high blood pressure, as compared to the non-severe group.

## Severity risk score

To select the best predictors, we performed stepwise model selection by AIC and built a logistic regression model based on a training data set of 163 patients selected by random split (S1 Fig, S2 Table, materials and methods). Seven predictors were selected: obesity, cardiovascular conditions distinct from high blood pressure, albumin, natremia, ferritin, CK and LDH (Table 2). Individual predictive performance measured by AUROC for four individual predictors (S3 Fig), ranked between 0.62 (natremia) and 0.83 (albumin). Albumin was the best individual predictor, with the following performance calculated with a cut-off of 26.95 g/L: accuracy 0.77 (95% Confidence Interval (CI) 0.70–0.83), sensitivity 0.66, specificity 0.85, PPV and NPV 0.77 and 0.76, respectively (S4 Fig).

To improve prediction performance a Covichem severity scoring was derived from the fitted logistic regression model. The AUROC was 0.91 (Fig 2A), the sensitivity and specificity were 0.85 and 0.88, respectively, which were better performance than albumin alone. The PPV and NPV were 0.85 and 0.88, respectively (Fig 2B). Overall, the prediction accuracy was 0.87 (95% CI 0.80–0.91). Predictor error was estimated with Leave One Out cross validation. The accuracy of 0.83 suggested that the model accuracy was not overly overestimated.

A test set of 40 patients (20 severe, S2 Table) reached similar performance with an AUROC of 0.93 and an accuracy of 0.83 (95% CI 0.67–0.93). Sensitivity was 0.80, specificity 0.85, PPV 0.84 and NPV 0.81 (Fig 2C).

Data from an independent cohort of 100 patients hospitalized at Pitié-Salpêtrière hospital (AP-HP, Paris) were collected to evaluate the Covichem severity score (S1 Fig, S2 Table). Performance on the external validation set were comparable to the internal validation set with an accuracy of 0.92 (95% CI 0.85–0.97), sensitivity of 0.89, specificity of 0.95, PPV of 0.93 and NPV of 0.91 (Fig 2D).

## Discussion

This study identified 2 clinical and 5 biochemical parameters as valuable predictors to build a COVID-19 severity score at patient's hospital admission. Overall, the characteristics of our discovery cohort are consistent with previous observations [15]. In particular, and in agreement

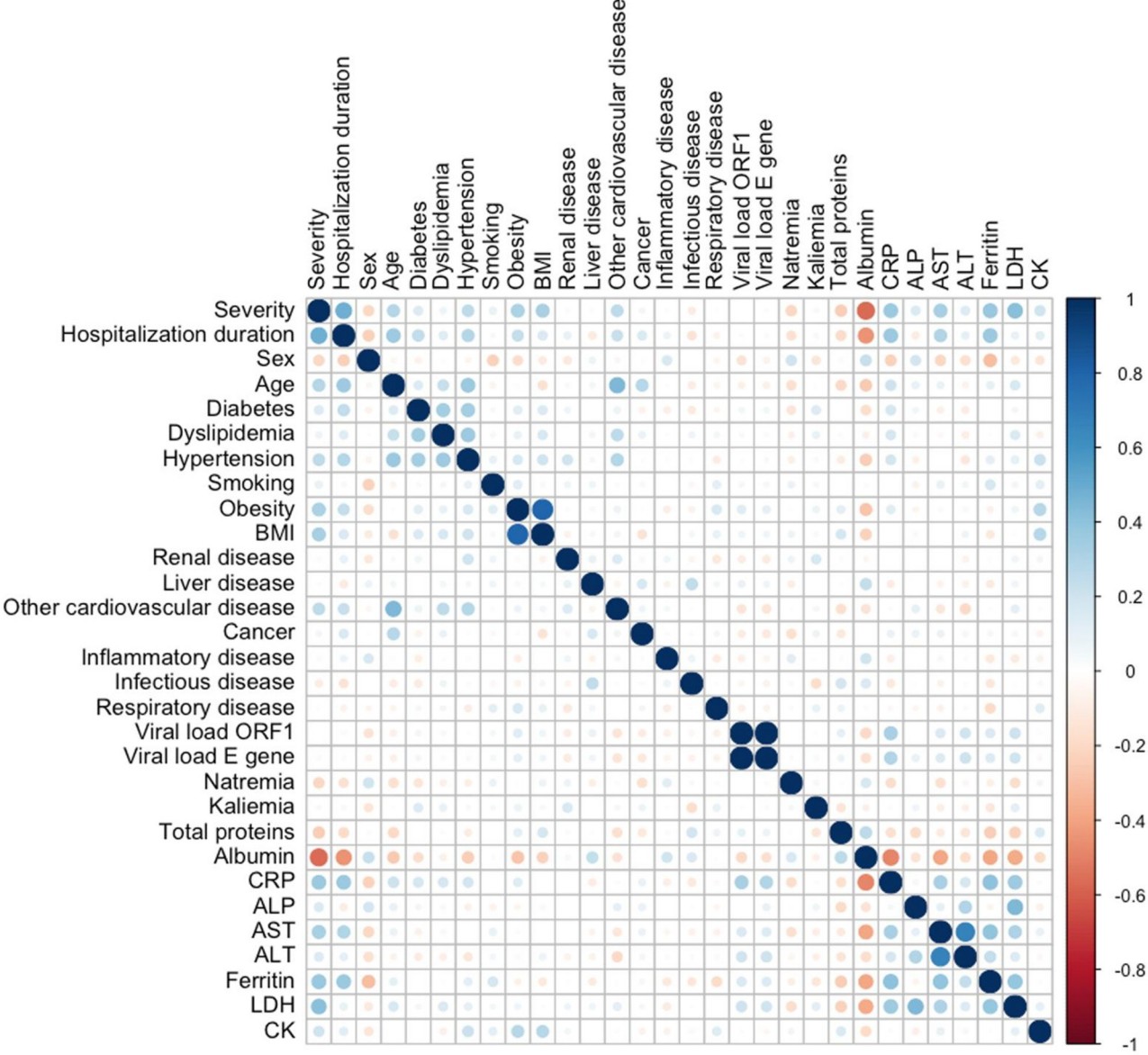

**Fig 1. Correlogram figuring out the relationship between each pair of clinical or biological variables in COVID-19 patients.** Positive correlations are displayed in blue and negative correlations in red. Color intensity and size of the circles are proportional to the correlation coefficients. On the right side of the correlogram, the legend color shows the correlation coefficients and the corresponding colors. ALP, Alkaline Phosphatase, AST, Aspartate Aminotransferase; ALT, Alanine Aminotransferase; BMI, Body Mass Index; CK; Creatine Kinase; CRP, C-reactive protein; LDH, Lactate Dehydrogenase.

with recent data [2, 4, 16], plasma albumin was the biochemical marker most strongly affected by COVID-19 severity, which caught our attention. In regular situations, hypoalbuminemia is a well-defined marker of malnutrition [17]. In urgent care, previous studies have shown that hypoalbuminemia at admission was associated with increased mortality in hospital medical emergency admission [18]. Patients below 27.4g/L (total cohort >20000 patients) presented a 30-day mortality of 31.7%. Odds ratios of death were over 3 times greater than in those with normal albumin levels. The predictive power on mortality of low albumin levels in an unselected acutely admitted medical population (5894 adult patients) found an OR of 3.91 when

**Table 2. Results of the stepwise model selection by Akaike Information Criterion (AIC) for model prediction of COVID-19 severity.**

| Predictors | Estimate | Standard error | z value | p value |
|---|---|---|---|---|
| (Intercept) | 14.6509961 | 8.5549991 | 1.713 | . |
| Obesity | 1.2903766 | 0.5205244 | 2.479 | * |
| Cardiovascular disease | 1.5237137 | 0.4843693 | 3.146 | ** |
| Natremia | -0.0961755 | 0.0619030 | -1.554 | 0.120268 |
| Albumin | -0.1805453 | 0.0473210 | -3.815 | *** |
| Ferritin | 0.0008822 | 0.0002885 | 3.058 | ** |
| LDH | 0.0033067 | 0.0019564 | 1.690 | . |
| CK | 0.0020180 | 0.0009898 | 2.039 | * |

*** p <0.001,

** p <0.01,

* p <0.05,

p <0.1. CK, Creatine Kinase, LDH, Lactate Dehydrogenase.

adjusting for CRP, liver disease, renal disease, cancer and rheumatologic disease [19]. Moreover, low albumin levels were observed in patients requiring intensive respiratory or vasopressor support during influenza H1N1 viral infection [20], with a cut-off value of 27g/L. The cut-off value of albumin observed here (26.95 g/L) is fully in agreement with these published data. Similar to our results, sensitivity was 0.79 and specificity 0.86 ([20], 0.66 and 0.85 in our study). Thus, it seems that distinct respiratory viral infections impact non-specific biochemical biomarkers in the same way, suggesting that they share systemic effects. Previous studies found that hypoalbuminemia was predictive for respiratory failure in MERS-CoV [21], and it may be considered that albumin drop is linked to liver failure [22]. However, our patients showed very mild if none liver enzymes increased levels, and considering that SARS-Cov-2 carries the potential to infect endothelial cells through their ACE-2 receptor, we believe possible that serum albumin level drops as a consequence of endotheliitis [23]. This hypothesis is in agreement with the fact that serum albumin is known as a biomarker of vascular permeability [24].

We built the Covichem score with unbiased machine learning modeling. The performance was better than any tested individual clinical or biochemical predictor. Of interest, the score included clinical parameters, the most relevant being obesity and cardiovascular comorbidities, as already described [25]. These latter conditions may worsen the drop in albuminemia observed in the severe disease, since low serum albumin levels are independently linked to several cardiovascular diseases [26]. Of note, high blood pressure, which was presumed of high risk of severity was not selected, in agreement with published data [27]. LDH and CK positively correlated with severity and were included in the score, which is not surprising and in agreement with recent data [28], considering the extended lung and possibly other tissues lesions induced by the infection. Even if we did not observe frank hyponatremia, the inclusion of this parameter in the score relates to low sodium plasma levels in severe infections [29]. Unlike Covichem, a severity score applied to COVID-19 identified CRP as a good predictor [30]. Instead, we found that ferritin was a better predictor. This may relate to the cytokine storm syndrome accompanying severe COVID-19, better reflected by strong hyperferritinemia than increased CRP [31]. Unlike other studies [25, 32] finding that uncontrolled diabetes was of more risk of severity, diabetes was not identified as a predictor in our study, possibly because of the low proportion of patients with this co-morbidity (19%, 39 patients). We did not include in our analysis sub-groups of diabetic patients (controlled and uncontrolled).

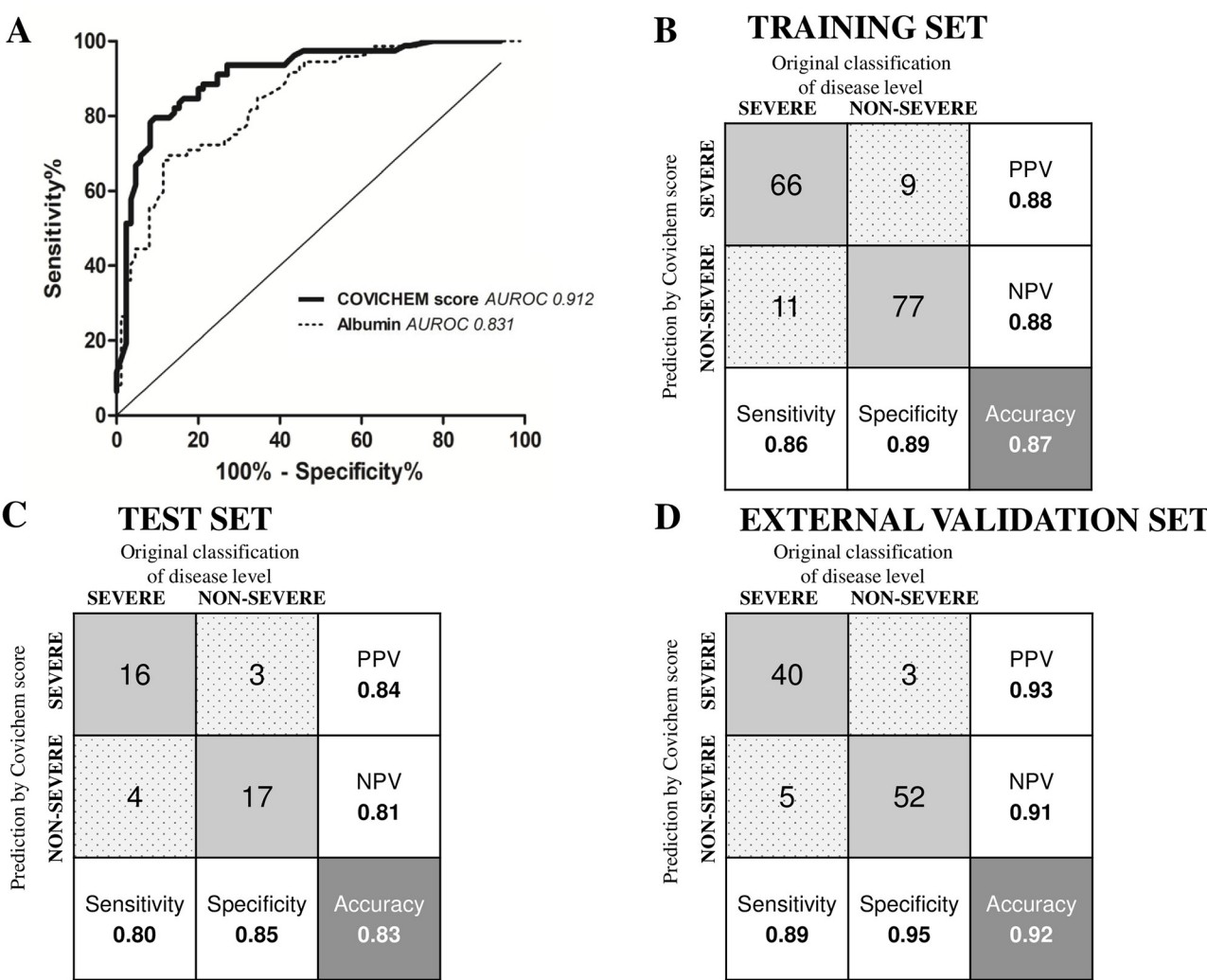

**Fig 2. Covichem score performances.** (A) Receiver Operating Characteristic curves for Covichem score and Albumin in predicting COVID-19 severity. The areas under Receiver Operating Characteristic curves (AUROC) are indicated on the graph legend. Confusion matrix and performance for Covichem score in the training (B), test (C) and external validation sets (D). Grey squares correspond to true positive and negative values and spotted grey squares represent false positive and negative values. Predictions were calculated for a cut-off of Covichem score at 0.5. NPV, Negative Predictive Value; PPV, Positive Predictive Value.

Finally, a sex ratio difference, males being more prone to severe disease than females, was found, as reported by others [33, 34], but was not selected in the model.

Importantly, Covichem showed very good performance on the independent cohort of patients from Paris, even if the labs used distinct methods of biochemical parameter measurements. The model is simple to set up since it necessitates only 7 variables. Moreover, the model is highly interpretable, since it is linear and the effects of the predictors are reflected by the regression coefficients. Compared to the N/L*CRP*D-dimer product [30], Covichem displayed better sensitivity and similar specificity (70 and 90% versus 86% and 89%). Covichem could be compared to other recently published scores [7, 8]. Gong et al. study is comparable to our work. The cohorts are of equivalent sizes (189 patients for the discovery cohort) and the selected 7 variables included albumin and LDH. However, variables were used in a nomogram, which needs skilled operators [8]. By contrast Covichem can be calculated with an Android©

APP (provided upon demand). The COVID-GRAM score is accessible online and needs imaging data or a long list of co-morbidities and clinical features. Our score displaying a slightly best AUC (0.91 versus 0.88), uses 30% less variables (7 versus 10) [35]. Our score could also be proposed for online calculation, but we think that the APP format is better since available anytime anywhere. Finally, the CALL score accuracy for our cohort was 67%, which is less than ours [7].

As stated above, our score presents numerous advantages: it is non-invasive, it does not need clinical exam by a medical doctor or imaging and could be performed by another health professional, it can be determined outside the hospital (online medical interview with lab prescription), it is independent of viral load determination, which can be easily saturable in pandemic conditions. Moreover, the test is cheap, fast (most labs will deliver results in <2h). The markers are available in all the routine labs around the world (no need of very specific markers, with limited access).

We believe that the Covichem score could be easily determined in patients who do not require hospitalization. It would be interesting to see if it detects the patients who are going to need hospitalization, as early as when they visit their general practitioner or when they present the first symptoms. In a pandemic context, such a tool could help family doctors. In line with lack of albumin normalization in COVID-19 patients with no improvement [15], it would be worthwhile to see how the Covichem score value evolves during the hospitalization, and its capacity to predict patient improvement. As we gain longer-term knowledge on post COVID-19 related disorders [36], it would be interesting to test the Covichem at admission as a predictive tool for long-term disease or to measure it distantly from hospital release as a follow-up marker of disease persistence.

We identified limitations of our study. First, we included a low number of patients. This study was conducted during the first wave in France, with a limited epidemic in Bordeaux area. Second, our study is retrospective, and we did not know at first what would be the best predictors and some data were missing. Third, we did not include racial/ethnicities as a variable although it may be a risk factor of severity [37]. This information is not systematically recorded in France, by law. At the technical level, we performed albumin measurement using the bromocresol purple dye, which accurately determines albumin levels in low ranges [24]. This point necessitates attention before using the Covichem score, especially by numerous labs using the bromocresol green dye that overestimates low albumin concentrations. This might be the reason why others have not included albuminemia in their scores. Noteworthy, albumin should be determined with bromocresol purple or by immune-based tests such as nephelemetry or turbidimetry.

## Conclusion

This study repositions certain biochemical analytes as relevant biomarkers of disease severity assessment in the COVID-19. This scoring system may help fast clinical decision with a simple blood test and co-morbidity assessment by anamnesis. It does not need deep exploration necessitating saturable equipments such as imaging, or procedures difficult to implement such as blood gas measures, necessitating trained medical gesture and strict pre-analytical conditions. It is implementable anywhere, including developing nations. Although the score needs external validation in multi-ethnic populations, the needed parameters are independent of race or ethnicity, which may presume the validity of the score worldwide.

## Supporting information

**S1 Fig. Flow chart of participants and distribution of COVID-19 severity in the discovery cohort (A) and in the external validation cohort (B).**
(TIF)

**S2 Fig. Density plots representation of continuous variables distribution between non severe and severe COVID-19 patients.** ALP, Alkaline Phosphatase, AST, Aspartate Aminotransferase; ALT, Alanine Aminotransferase; BMI, Body Mass Index; CK; Creatine Kinase; CRP, C-reactive protein; LDH, Lactate Dehydrogenase.
(TIF)

**S3 Fig. Comparison of Receiver Operating Characteristic curves for clinical and biological variables in predicting COVID-19 severity.** The areas under Receiver Operating Characteristic curves (AUROC) are indicated for each variable on the graph legend. BMI, Body Mass Index; LDH, Lactate Dehydrogenase.
(TIF)

**S4 Fig. Confusion matrix and performance of Albumin level in predicting COVID-19 severity.** Grey squares correspond to true positive and true negative values, spotted grey squares represent false positive and false negative values. Predictions were calculated for a cut-off of albumin at 26.95 g/L. NPV, Negative Predictive Value; PPV, Positive Predictive Value.
(TIF)

**S1 Table. Pearson's correlation coefficients for each variable with COVID-19 severity.**
ALP, Alkaline Phosphatase; ALT, Alanine Aminotransferase; AST, Aspartate Aminotransferase; BMI, Body Mass Index; CK, Creatine Kinase; CRP, C-reactive protein; LDH, Lactate Dehydrogenase.
(PDF)

**S2 Table. Baseline characteristics of the training, test and external validation sets.** Continuous variables are expressed as median (25th, 75th percentile). Discrete variables are presented as absolute (relative) frequencies of patients. ALP, Alkaline Phosphatase; ALT, Alanine Aminotransferase; BMI, Body Mass Index; CK, Creatine Kinase; CRP, C-reactive protein; AST, Aspartate Aminotransferase; Ct, Cycle threshold; LDH, Lactate Dehydrogenase; NA, not available.
(PDF)

**S1 File.**
(XLSX)

## Acknowledgments

We thank Pr Dominique Bonnefont-Rousselot for giving us the opportunity to include the parisian cohort in study, Pauline Ratouit and Hedy Nemeur for data and samples collection, Dr Annie Bérard and Pr Marie-Edith Lafon for their critical reading of the manuscript. The authors acknowledge the Bordeaux University Hospital Biobank "CRB- Bordeaux Biothèques Santé", BB-0033-00094 for technical assistance in managing patient sample storage. We are very grateful to Dr Jérémie Bureau for his methodological support and his strong expertise in statistical analysis.

## Author Contributions

**Conceptualization:** Marie-Lise Bats, Benoit Rucheton, Tara Fleur, Arthur Orieux, Clément Chemin, Sébastien Rubin, Brigitte Colombies, Arnaud Desclaux, Claire Rivoisy, Etienne Mériglier, Etienne Rivière, Alexandre Boyer, Didier Gruson, Isabelle Pellegrin, Pascale Trimoulet, Isabelle Garrigue, Rana Alkouri, Charles Dupin, François Moreau-Gaudry, Aurélie Bedel, Sandrine Dabernat.

**Data curation:** Marie-Lise Bats, Benoit Rucheton, Tara Fleur, Arthur Orieux, Clément Chemin, Brigitte Colombies, Arnaud Desclaux, Claire Rivoisy, Etienne Mériglier, Isabelle Pellegrin, Pascale Trimoulet, Isabelle Garrigue, Rana Alkouri, François Moreau-Gaudry, Sandrine Dabernat.

**Formal analysis:** Marie-Lise Bats, Arnaud Desclaux, Aurélie Bedel, Sandrine Dabernat.

**Methodology:** Marie-Lise Bats, Benoit Rucheton, Arthur Orieux, Clément Chemin, Sébastien Rubin, Arnaud Desclaux, Etienne Rivière, Alexandre Boyer, Didier Gruson, Aurélie Bedel, Sandrine Dabernat.

**Project administration:** Sandrine Dabernat.

**Resources:** Benoit Rucheton, Isabelle Pellegrin.

**Software:** Charles Dupin.

**Supervision:** Sandrine Dabernat.

**Validation:** Marie-Lise Bats, Sébastien Rubin, Brigitte Colombies, Aurélie Bedel, Sandrine Dabernat.

**Visualization:** Marie-Lise Bats, Benoit Rucheton, Tara Fleur, Arthur Orieux, Clément Chemin, Sébastien Rubin, Brigitte Colombies, Claire Rivoisy, Etienne Mériglier, Etienne Rivière, Alexandre Boyer, Didier Gruson, Isabelle Pellegrin, Pascale Trimoulet, Isabelle Garrigue, Rana Alkouri, Charles Dupin, François Moreau-Gaudry, Aurélie Bedel, Sandrine Dabernat.

**Writing – original draft:** Marie-Lise Bats, Sandrine Dabernat.

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
