## [Decision Letter · Decision Letter 0]

15 Mar 2021

PONE-D-20-38092

Covichem: a biochemical severity risk score of COVID-19

upon hospital admission

PLOS ONE

Dear Dr. Dabernat,

Thank you for submitting your manuscript to PLOS ONE. After careful consideration, we feel that it has merit but does not fully meet PLOS ONE’s publication criteria as it currently stands. Therefore, we invite you to submit a revised version of the manuscript that addresses the points raised during the review process.

The paper is quite interesting and well written. However, the conclusions should be re written and focused on results. 

We look forward to receiving your revised manuscript.

Kind regards,

Chiara Lazzeri

Academic Editor

PLOS ONE

Journal Requirements:

Reviewers' comments:

Reviewer's Responses to Questions

**Comments to the Author**

1. Is the manuscript technically sound, and do the data support the conclusions?

Reviewer #1: Partly

2. Has the statistical analysis been performed appropriately and rigorously? 

Reviewer #1: Yes

3. Have the authors made all data underlying the findings in their manuscript fully available?

Reviewer #1: Yes

4. Is the manuscript presented in an intelligible fashion and written in standard English?

Reviewer #1: Yes

5. Review Comments to the Author

Reviewer #1: I think there is academic merit in this work but the contribution of the tool to clinicians or other stakeholders needs to be clearer. The conclusion seemed to focus on something different than what was presented as the aim of the paper.

Introduction

Line 52 -55: It is not clear what “soon after the first phase of COVID-19” refer to, is this first phase in France? what timeline?. Additionally, “few” parameters seem inaccurate when taking into account that Rod et al. 2020 reported 60 risk factors for disease severity using early studies coming from China and Singapore with at least 50% of those been biomarkers that could be tested in blood.

Line 68 – 69: The following sentence was not fully clear to me “Turn-around-time is short to obtain automated biochemical parameters 70 measures (<2h)”.

I would suggest to double check English and the connectors between sentences in the last paragraph of the introduction. The authors seem to follow the convention of writing the aim of the research at the end of the introduction. However, the last sentence sound more like an action/task that belongs to the methods section and it does not appear to me to be written as an aim.

Methods

Line 122-123: It is not fully clear what “referring” to odds ratios means. Additionally, odds ratios are a different mathematical construct than probability ratios (which are commonly called risk ratios under an epidemiological and the quantitative notions of risk). Generally in epidemiology both odds ratios and risk ratios are called measures of association between variables, it might be worthwhile to consider the use the term “measures of association”. This would likely make a value free statement in the results section and leave value judgements about risk for the discussion and the conclusion.

Results

Table 1: There is a risk that the use of odds ratios overestimates relative risks when the rare disease assumption (prevalence less than 10%) is not present. This seems to happen for most comorbidities in Table 1. This could mean that the presented odds estimates might be overestimated. If the interest is risk, why use logistic regression instead of other generalized linear models (e.g. log binomial model) that offer relative risk as the output and not odds ratios?.

Discussion

Line 300 – What machine learning model? I could not find any machine learning model in the methods section.

Should there be a table comparing existing predictors against the proposed model?

I think the discussion should be supplemented with discussions of other papers that are addressing the topic. Eg.

Drager, L. F., Pio-Abreu, A., Lopes, R. D., & Bortolotto, L. A. (2020). Is Hypertension a Real Risk Factor for Poor Prognosis in the COVID-19 Pandemic? Curr Hypertens Rep, 22(6), 43. doi:10.1007/s11906-020-01057-x

Gebhard, C., Regitz-Zagrosek, V., Neuhauser, H. K., Morgan, R., & Klein, S. L. (2020). Impact of sex and gender on COVID-19 outcomes in Europe. Biol Sex Differ, 11(1), 29. doi:10.1186/s13293-020-00304-9

Leung, C. (2020). Risk factors for predicting mortality in elderly patients with COVID-19: A review of clinical data in China. Mechanisms of Ageing and Development, 188, 111255. doi:https://doi.org/10.1016/j.mad.2020.111255

Rod, J. E., Oviedo-Trespalacios, O., & Cortes-Ramirez, J. (2020). A brief-review of the risk factors for covid-19 severity. Rev Saude Publica, 54, 60. doi:10.11606/s1518-8787.2020054002481

Singh, A. K., & Khunti, K. (2020). Assessment of risk, severity, mortality, glycemic control and antidiabetic agents in patients with diabetes and COVID-19: A narrative review. Diabetes Res Clin Pract, 165, 108266. doi:10.1016/j.diabres.2020.108266

Wolff, D., Nee, S., Hickey, N. S., & Marschollek, M. (2020). Risk factors for Covid-19 severity and fatality: a structured literature review. Infection, 1-14. doi:10.1007/s15010-020-01509-1

Zheng, Z., Peng, F., Xu, B., Zhao, J., Liu, H., Peng, J., . . . Tang, W. (2020). Risk factors of critical & mortal COVID-19 cases: A systematic literature review and meta-analysis. J Infect, 81(2), e16-e25. doi:10.1016/j.jinf.2020.04.021

Limitations

There should be a discussion regarding the external validity of the score beyond the used sample sizes in the particular populations that were used to propose and validate. Why should other clinicians in Europe/ developed nations expect similar results?. What about developing nations?.

Conclusions

Lines 359 – 361: Rod et al. 2020 review of early studies coming from china looking at risk factors for a composite index of severe-fatal the COVID-19 disease propose albumin as one of the most consistent risk factors for severe-fatal COVID-19. Based on this information, it is hard to understand how the present paper is “repositioning” albumin as a standard of care of any COVID-19 disease patient admitted to urgent care. Please be more specific what is the contribution that the present paper is adding to the literature in regards albumin levels.

The conclusion needs a lot of work. I found hard to find the logic between the aim being to propose, validate and promote the use of a score by using an app and to conclude that the paper has reposition albumin as an important risk factor (which is not accurate based on the above-mentioned rationale).

What future research directions does the current state of development of this score lead researchers to?.

6. PLOS authors have the option to publish the peer review history of their article (what does this mean?). If published, this will include your full peer review and any attached files.

Reviewer #1: No

---

## [Author Response · Author response to Decision Letter 0]

9 Apr 2021

We would like to thank the reviewer and the editor for their fair review of our work.

Line 52 -55: It is not clear what “soon after the first phase of COVID-19” refer to, is this first phase in France? what timeline?

We now specify the timeline as being April 2020.

Additionally, “few” parameters seem inaccurate when taking into account that Rod et al. 2020 reported 60 risk factors for disease severity using early studies coming from China and Singapore with at least 50% of those been biomarkers that could be tested in blood.

We agree that several (even now numerous) studies have described risk factors for COVID-19 severity. However, when undertaking the present study, our analysis of the literature, focused to biochemical parameters, showed that only a few were consistently found linked to disease severity and not to other outcomes (such as death ratio, or hospitalization rates). We thank the reviewer for providing the “Rod, J. E., Oviedo-Trespalacios, O., & Cortes-Ramirez, J. (2020). A brief-review of the risk factors for covid-19 severity. Rev Saude Publica, 54, 60. doi:10.11606/s1518-8787.2020054002481” reference, that we now cite, and which summarizes the available data on COVID severity in April 2020. As shown in the Table of this review, CRP, D-Dimer and Albumin show consistency between studies. We changed the introduction of the manuscript accordingly, and cite Rod et al.

Line 68 – 69: The following sentence was not fully clear to me “Turn-around-time is short to obtain automated biochemical parameters 70 measures (<2h)”. I would suggest to double check English and the connectors between sentences in the last paragraph of the introduction. The authors seem to follow the convention of writing the aim of the research at the end of the introduction. However, the last sentence sound more like an action/task that belongs to the methods section and it does not appear to me to be written as an aim.

We agree with the reviewer’s comment, and we now limited the last sentence of the introduction to the aim of the study, which was to identify biochemical parameters useful to establish a COVID-19 severity score.

Line 122-123: It is not fully clear what “referring” to odds ratios means. Additionally, odds ratios are a different mathematical construct than probability ratios (which are commonly called risk ratios under an epidemiological and the quantitative notions of risk). Generally in epidemiology both odds ratios and risk ratios are called measures of association between variables, it might be worthwhile to consider the use the term “measures of association”. This would likely make a value free statement in the results section and leave value judgements about risk for the discussion and the conclusion.

Table 1: There is a risk that the use of odds ratios overestimates relative risks when the rare disease assumption (prevalence less than 10%) is not present. This seems to happen for most comorbidities in Table 1. This could mean that the presented odds estimates might be overestimated. If the interest is risk, why use logistic regression instead of other generalized linear models (e.g. log binomial model) that offer relative risk as the output and not odds ratios?.

The reviewer is right to underline that according to the mathematical calculation of odd ratios, there is a risk of overestimation of the studied effects. We originally provided these data as supplementary data, because odd ratios are commonly provided in such studies. However, we did not determine the score based on these results and propose to withdraw supplemental Table S2 as the part in text referring to this Table and odd ratios.

Line 300 – What machine learning model? I could not find any machine learning model in the methods section.

It is specified in the M&M and in the result sections: we used a stepwise model selection by Akaike Information Criterion.

Should there be a table comparing existing predictors against the proposed model?

We did compare the best single predictors, found by others over the literature, with the proposed score by building ROC and calculating AUROC (Figure S3). There are no published ‘”existing single predictors”. In addition, as mentioned in the discussion we compared our score to the “CALL” score.

I think the discussion should be supplemented with discussions of other papers that are addressing the topic. Eg.

All the mentioned studies are now cited in the discussion.

There should be a discussion regarding the external validity of the score beyond the used sample sizes in the particular populations that were used to propose and validate. Why should other clinicians in Europe/ developed nations expect similar results?. What about developing nations?.

We provide this in the conclusion.

Lines 359 – 361: Rod et al. 2020 review of early studies coming from china looking at risk factors for a composite index of severe-fatal the COVID-19 disease propose albumin as one of the most consistent risk factors for severe-fatal COVID-19. Based on this information, it is hard to understand how the present paper is “repositioning” albumin as a standard of care of any COVID-19 disease patient admitted to urgent care. Please be more specific what is the contribution that the present paper is adding to the literature in regards albumin levels. The conclusion needs a lot of work. I found hard to find the logic between the aim being to propose, validate and promote the use of a score by using an app and to conclude that the paper has reposition albumin as an important risk factor (which is not accurate based on the above-mentioned rationale). What future research directions does the current state of development of this score lead researchers to?

We agree with the reviewer that mentioning albumin at this stage of the manuscript was not appropriate. The sentence regarding albumin is now removed. We just kept the discussion part about albumin in the discussion section.

We thank the reviewer for suggesting to open the study to future directions. We now provide a conclusion in the context of the disease evolution with the new variants.

---

## [Editor Report · Decision Letter 1]

19 Apr 2021

Covichem: a biochemical severity risk score of COVID-19 upon hospital admission

PONE-D-20-38092R1

Dear Dr. Dabernat,

We’re pleased to inform you that your manuscript has been judged scientifically suitable for publication and will be formally accepted for publication once it meets all outstanding technical requirements.

Kind regards,

Chiara Lazzeri

Academic Editor

PLOS ONE
---

## [Editor Report · Acceptance letter]

27 Apr 2021

PONE-D-20-38092R1 

Covichem: a biochemical severity risk score of COVID-19 upon hospital admission 

Dear Dr. Dabernat:

I'm pleased to inform you that your manuscript has been deemed suitable for publication in PLOS ONE. Congratulations! Your manuscript is now with our production department. 

Kind regards, 

on behalf of

Dr. Chiara Lazzeri 

Academic Editor

PLOS ONE